# TOMA: Topological Map Abstraction for Reinforcement Learning

## Abstract

Animals are able to discover the topological map (graph) of surrounding environment, which will be used for navigation. Inspired by this biological phenomenon, researchers have recently proposed to *learn* a graph representation for Markov decision process (MDP) and use such graphs for planning in reinforcement learning (RL). However, existing learning-based graph generation methods suffer from many drawbacks. One drawback is that existing methods do not learn an abstraction for graphs, which results in high memory and computation cost. This drawback also makes generated graph non-robust, which degrades the planning performance. Another drawback is that existing methods cannot be used for facilitating exploration which is important in RL. In this paper, we propose a new method, called topological map abstraction (TOMA), for learning-based graph generation. TOMA can learn an abstract graph representation for MDP, which costs much less memory and computation cost than existing methods. Furthermore, TOMA can be used for facilitating exploration. In particular, we propose *planning to explore*, in which TOMA is used to accelerate exploration by guiding the agent towards unexplored states. A novel experience replay module called *vertex memory* is also proposed to improve exploration performance. Experimental results show that TOMA can outperform existing methods to achieve the state-of-the-art performance.

## 1 Introduction

Animals are able to discover topological map (graph) of surrounding environment (O'Keefe and Dostrovsky, 1971; Moser et al., 2008), which will be used as hints for navigation. For example, previous maze experiments on rats (O'Keefe and Dostrovsky, 1971) reveal that rats can create mental representation of the maze and use such representation to reach the food placed in the maze. In cognitive science society, researchers summarize these discoveries in *cognitive map theory* (Tolman, 1948), which states that animals can extract and code the structure of environment in a compact and abstract map representation.

Inspired by such biological phenomenon, researchers have proposed to generate topological graph representation for Markov decision process (MDP) and use such graphs for planning in reinforcement learning (RL). Early graph generation methods (Mannor et al., 2004) are usually *prior-based*, which apply some human prior to aggregate similar states to generate vertices. Recently, researchers propose some *learning-based* graph generation algorithms which learn such state aggregation automatically. Such methods have been proved to be better than prior-based methods (Metzen, 2013). These methods generally treat the states in a replay buffer as vertices. For the edges of the graphs, some methods like SPTM (Savinov et al., 2018) train a reachability predictor via self-supervised learning and combine it with human experience to construct the edges. Other methods like SoRB (Eysenbach et al., 2019) exploit a goal-conditioned agent to estimate the distance between vertices, based on which edges are constructed. These existing methods suffer from the following drawbacks. Firstly, these methods do not learn an abstraction for graphs and usually consider all the states in the buffer as vertices (Savinov et al., 2018), which results in high memory and computation cost. This drawback also makes generated graph non-robust, which will degrade the planning performance. Secondly, existing methods cannot be used for facilitating exploration, which is important in RL. In particular, methods like SPTM rely on human sampled trajectories to generate the graph, which is infeasible in RL exploration. Methods like SoRB require training another goal-conditioned agent. Such training

procedure assumes knowledge of the environment since it requires to generate several goal-reaching tasks to train the agent. This practice is also intractable in RL exploration.

In this paper, we propose a new method, called TOpological Map Abstraction (TOMA), for learning-based graph generation. The main contributions of this paper are outlined as follows:

- TOMA can learn to generate an abstract graph representation for MDP. Different from existing methods in which each vertex of the graph represents a state, each vertex in TOMA represents a cluster of states. As a result, compared with existing methods TOMA has much less memory and computation cost, and can generate more robust graph for planning.

- TOMA can be used to facilitate exploration. In particular, we propose *planning to explore*, in which TOMA is used to accelerate exploration by guiding the agent towards unexplored states. A novel experience replay module called *vertex memory* is also proposed to improve exploration performance.

- Experimental results show that TOMA can robustly generate abstract graph representation on several 2D world environments with different types of observation and can outperform previous learning-based graph generation methods to achieve the state-of-the-art performance.

## 2 ALGORITHM

### 2.1 NOTATIONS

In this paper, we model a RL problem as a Markov decision process (MDP). A MDP is a tuple $M(S, A, R, \gamma, P)$, where $S$ is the state space, $A$ is the action space, $R : S \times A \to \mathbb{R}$ is a reward function, $\gamma$ is a discount factor and $P(s_{t+1}|s_t, a_t)$ is the transition dynamic. $\rho(x, y) = \|x - y\|_2$ denotes Euclidean distance. $G(V, E)$ denotes a graph, where $V$ is its vertex set and $E$ is its edge set. For any set $X$, we define its indicator function $\mathbb{1}_X(x)$ as follows: $\mathbb{1}_X(x) = 1$ if $x \in X$, $\mathbb{1}_X(x) = 0$ if $x \notin X$.

### 2.2 TOMA

Figure 1 gives an illustration of TOMA, which tries to map states to an abstract graph. A landmark set $L$ is a subset of $S$ and each landmark $l_i \in L$ is a one-to-one correspondence to a vertex $v_i$ in the graph. Each $l_i$ and $v_i$ will represent a cluster of states. In order to decide which vertex a state $s \in S$ corresponds to, we first use a locality sensitive embedding function $\phi_\theta$ to calculate its latent representation $z = \phi_\theta(s)$ in the embedding space $Z$. Then if $z$'s nearest neighbor in the embedded landmark set $\phi_\theta(L) = \{\phi_\theta(l)| l \in L\}$ is $\phi_\theta(l_i)$, we will map $s$ to vertex $v_i \in V$.

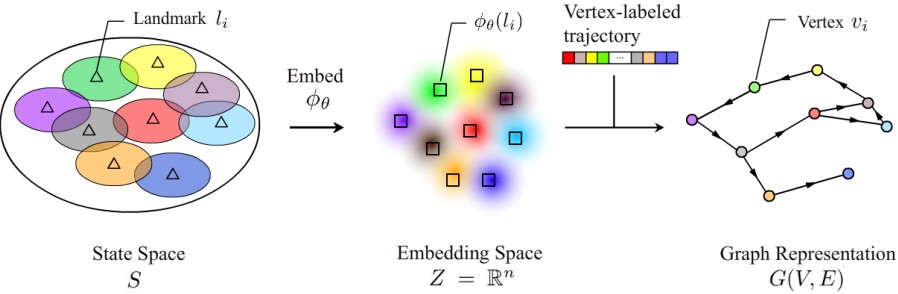

Figure 1: Illustration of TOMA. We will pick up some states as landmarks (colored triangles) in the state space of the original MDP $M$. Each landmark $l_i$ is a one-to-one correspondence to a vertex $v_i$ (colored circles) in graph $G$ and covers some areas in $S$. Embedding $\phi_\theta$ is trained by self-supervised learning. We will label each state on a trajectory with a corresponding vertex and use it to generate the graph dynamically.

### 2.2.1 LOCALITY SENSITIVE EMBEDDING

A locality sensitive embedding is a local distance preserving mapping $\phi_\theta$ from state space $S$ to an embedding space $Z$, which is an Euclidean space $\mathbb{R}^n$ in our implementation. Given a trajectory $T = (s_0, a_0, s_1, a_1, ....s_n)$, we can use $d_{ij} = |j - i|/r$ to estimate the distance between $s_i$ and $s_j$. Here $r$ is a radius hyper-parameter to re-scale the distance and we will further explain its meaning later. In practice, however, $d_{ij}$ is a noizy estimation for shortest distance and approximating it directly won't converge in most cases. Hence, we propose to estimate which interval the real distance lies in. First, we define three indicator functions:

$$\chi_1(x) = \mathbb{1}_{[0,1]}(x), \tag{1}$$

$$\chi_2(x) = \mathbb{1}_{(1,3]}(x), \tag{2}$$

$$\chi_3(x) = \mathbb{1}_{(3,+\infty)}(x), \tag{3}$$

which mark three disjoint regions $[0, 1], (1, 3], (3, +\infty)$, respectively. Then we define an anti-bump function $\xi_{a,b}(x) = \text{Relu}(-x + a) + \text{Relu}(x - b)$. Here, $\text{Relu}(x) = \max(0, x)$ is the rectified linear unit (ReLU) function (Glorot et al., 2011). With this $\xi(x)$, we can measure the deviation from the above intervals. Let

$$\mathcal{L}_1(x) = \xi_{-\infty,1}(x) = \text{Relu}(x - 1), \tag{4}$$

$$\mathcal{L}_2(x) = \xi_{1,3}(x) = \text{Relu}(-x + 1) + \text{Relu}(x - 3), \tag{5}$$

$$\mathcal{L}_3(x) = \xi_{3,+\infty}(x) = \text{Relu}(-x + 3), \tag{6}$$

and let $d'_{ij} = \rho(\phi_\theta(s_i), \phi_\theta(s_j))$ denote the distance between $s_i$ and $s_j$ in the embedding space. Our embedding loss is defined as

$$\mathcal{L}(\theta) = \mathbb{E}_{(s_i,s_j) \sim P_s} \left( \chi_1(d_{ij})\mathcal{L}_1(d'_{ij}) + \lambda_1 \chi_2(d_{ij})\mathcal{L}_2(d'_{ij}) + \lambda_2 \chi_3(d_{ij})\mathcal{L}_3(d'_{ij}) \right). \tag{7}$$

Here $P_s$ is a sample distribution which will be described later, $\lambda_1$ and $\lambda_2$ are two hyper-parameters to balance the importance of the estimation for different distances. We find that a good choice is to pick $\lambda_1 = 0.5, \lambda_2 = 0.2$ to ensure that our model focuses on the terms with lower variance. In this equation, there are some critical components to notice:

**Radius** $r$ As we will see later, the hyper-parameter $r$ will determine the granularity of each graph vertex, which we term as radius. If we define the $k$-ball neighborhood of $s \in S$ to be

$$B_k(s) = \{s' \in S | \rho(\phi_\theta(s'), \phi_\theta(s)) < k\}, \tag{8}$$

then $B_k(s)$ will cover more states when $r$ is larger. During the graph generation process, we will remove redundant vertices by checking whether $B_1(l_i)$ and $B_1(l_j)$ intersect too much. Re-scaling by $r$ makes it easier to train the embedding function.

**Sample Distribution** $P_s$ The state pair $(s_i, s_j)$ in the loss function is sampled from a neighborhood biased distribution $P_s$. We will sample $(s_i, s_j)$ $(i < j)$ with probability $\alpha$, if $j - i \le 4r$. And we will sample $(s_i, s_j)$ $(i < j)$ with probability $1 - \alpha$, if $j - i > 4r$. We simply take $\alpha = 0.5$ and the choice of $\alpha$ is not sensitive in our experiment. In the implementation, we use this sample distribution to draw samples from trajectory and put them into a replay pool. Then we train the embedding function by uniformly drawing samples from the pool.

**Anti-Bump Functions** The idea of anti-bump function is inspired by the *partition of unity theorem* in differential topology (Hirsch, 1997), where a bunch of bump functions are used to glue the local charts of manifold together so as to derive global properties of differential manifolds. In proofs of many differential topology theorems, one crucial step is to use bump function to segregate each local chart into three disjoint regions by radius 1, 2 and 3, which is analogous to our method. The loss function is crucial in our method, as in experiment we find that training won't converge if we replace this loss function with a commonly used $L_2$ loss.

### 2.2.2 DYNAMIC GRAPH GENERATION

An abstract graph representation $G(V, E)$ should satisfy the following basic requirements:

- *Simple*: For any $v_i, v_j \in V$, if $v_i \ne v_j$, $B_1(l_i) \cap B_1(l_j)$ should not contain too many elements.

- *Accurate*: For any $v_i, v_j \in V$ and $v_i \neq v_j$, $\langle v_i, v_j \rangle \in E$ if and only if the agent can travel from some $s \in B_1(l_i)$ to some $s' \in B_1(l_j)$ in a small number of steps.

- *Abundant*: $\cup_{i:v_i \in V} B_1(l_i)$ should cover the states as many as possible.

- *Dynamic*: $G$ grows dynamically, by absorbing topology information of novel states.

In the following content, we show a dynamic graph generation method fulfilling such requirements. First, we introduce the basic operations in our generation procedure. The operations can be reduced to the following three categories:

---

**Initializing**
**[I1: Initialize]** If $V = \emptyset$, we will pick a landmark from currently sampled trajectories and add a vertex into $V$ accordingly. In our implementation, this landmark is the initial state of the agent.

---

**Adding**
**[A1: Add Labels]** For each state $s$ on a trajectory, we label it with its corresponding graph vertex. Let $i = \arg\min_{j:v_j \in V} \rho(\phi_\theta(s), \phi_\theta(l_j))$ and $d = \rho(\phi_\theta(s), \phi_\theta(l_i))$. There are three possible cases: (1) $d \in [0, 1.5]$. We label $s$ with $v_i$. (2) $d \in [2, 3]$. We consider $s$ as an appropriate landmark candidate. Therefore, we label $s$ with NULL but add it to a candidate queue. (3) Otherwise, $s$ is simply labelled with NULL.
**[A2: Add Vertices]** We move some states from the candidate queue into the landmark set and update $V$ accordingly. Once a state is added to the landmark set, we will relabel it from NULL to its vertex identifier.
**[A3 Add Edges]** Let the labelled trajectory to be $(v_{i_0}, v_{i_1}, ..., v_{i_n})$. If we find $v_{i_k}$ and $v_{i_{k+1}}$ are different vertices in the existing graph, we will add an edge $\langle v_{i_k}, v_{i_{k+1}} \rangle$ into the graph.

---

**Checking**
**[C1: Check Vertices]** If $\rho(\phi_\theta(l_i), \phi_\theta(l_j)) < 1.5$, then we will merge $v_i$ and $v_j$.
**[C2: Check Edges]** For any edge $\langle v_i, v_j \rangle$, if $\rho(\phi_\theta(l_i), \phi_\theta(l_j)) > 3$, we will remove this edge.

---

For efficient nearest neighbor search, we use Kd-tree (Bentley, 1975) to manage the vertices. Based on the above operations, we can get our graph generation algorithm TOMA which is summarized in Algorithm 1.

---

**Algorithm 1** Topological Map Abstraction (TOMA)

---

1: Pool $P \leftarrow \emptyset$. Vertex set $V \leftarrow \emptyset$. Edge set $E \leftarrow \emptyset$. Graph $G(V, E)$.
2: **for** $t = 1, 2, ...$ **do**
3:     Sample a trajectory $T$ using some policy $\pi$ or by random.
4:     Sample state pairs from $T$ using distribution $P_s$ and put them to $P$.
5:     Training the embedding function $\phi_\theta$ using samples from $P$.
6:     Initialize $G$ using (I1) if it's empty.
7:     Add vertices and edges using (A1) to (A3).
8:     Check the graph using (C1) to (C2).
9: **end for**

---

### 2.2.3 INCREASING ROBUSTNESS

In practice, we find TOMA sometimes provides inaccurate estimation on image domains without rich visual information. This is similar to the findings of (Eysenbach et al., 2019), which uses an ensemble of distributional value function for robust distance estimation. To increase robustness, we can also use an ensemble of embedding functions to provide reliable neighborhood relationship estimation on these difficult domains. The functions in the ensemble are trained with data drawn from the same pool. During labelling, each function will vote a nearest neighbor for the given observation and TOMA will select the winner as the label. To evaluate the distance between states, we use the average of the distance estimation of all embedding functions. In (Eysenbach et al., 2019), the authors find that ensemble is an indispensable component in their value function based method for all applications. On the contrary, TOMA does not require ensemble to increase robustness on applications with rich information.

## 2.3 PLANNING TO EXPLORE

Since the graph of TOMA expands dynamically as agent samples in the environment, it can be fitted into standard RL settings to facilitate exploration. We choose the furthest vertex or the least visited vertex as the ultimate goal for agent in each episode. During sampling we periodically run Dijkstra's algorithm to figure out the path towards the goal from the current state, and the vertices on the path are used as intermediate goals. To ensure that the agent can stably reach the border, we further introduce the following memory module.

**Vertex Memory**   We observe that the agent usually fails to explore efficiently simply because it forgets how to reach the border of explored area as training goes on. In order to make agent recall the way to the border, we require that each vertex $v_i$ should maintain a small replay buffer to record successful transitions into the cluster of $v_i$. Then, if our agent is going towards goal $g$ and the vertices on the shortest path towards the corresponding landmark of $g$ are $v_1, v_2, ..., v_k$, then we will draw some experience from the replay pool of $v_1, v_2,..., v_k$ to inform the agent of relevant knowledge during training. In the implementation, we use the following replay strategy: half of the training data are drawn from experience of vertex memory which provides task-specific knowledge, while the other half are drawn from normal hindsight experience replay (HER) (Andrychowicz et al., 2017) which provides general knowledge. We will use the sampled trajectory to update the memory of visited vertices at the end of each epoch. The overall procedure is summarized in Algorithm 2.

---

**Algorithm 2** Planning to Explore with TOMA

---
1:  **for** $t = 1, 2, ...$ **do**
2:      Set a goal $g$ using some criterion.
3:      Sample a trajectory $T$ under the guidance of intermediate goals.
4:      Update graph using $T$ (Algorithm 1).
5:      Update vertex memory and HER using $T$.
6:      Train the policy $\pi$ using experience drawn from vertex memory and HER.
7:  **end for**

---

## 3 EXPERIMENTS

In the experiments, we first show that TOMA can generate abstract graphs via visualization and demonstrate that such graph is suitable for planning. Then we carry out exploration experiment in some sparse reward environments and show that TOMA can facilitate exploration.

### 3.1 GRAPH GENERATION

#### 3.1.1 VISUALIZATION

In this section, we test whether TOMA can generate abstract graph via visualization. In order to provide intuitive visualization, we use several 2D world environments to test TOMA, which are shown in Figure 2(a). The scale of these planar worlds is $100 \times 100$. In these worlds, there are some walls which agent can not cross through. The agent can take 4 different actions at each step: going up, down, left or right for one unit distance. To simulate various reinforcement learning domains, we test the agent on four different types of observation respectively: sensor, noisy sensor, MNIST digit (LeCun and Cortes, 2010) and top-down observation. Sensor observation is simply the $(x, y)$ coordinates of the agent and the noizy sensor observation is the coordinates with 8 random features. Both MNIST digit and top-down observation are image observations. MNIST digit observation is a mixture of MNIST digit images, which is similar to the reconstructed MNIST digit image of variational auto-encoders (Kingma and Welling, 2014). The observed digit is based on the agent's position and varies continuously as agent moves in this world. Top-down observation is a blank image with a white dot indicating the agent's location. We use three different maps: "Empty", "Lines" and "Four rooms". Since in this experiment we only care about whether TOMA can generate abstract graph representation from enough samples, we spawn a random agent at a random position in the map at the beginning of each episode. Each episode lasts 1000 steps and we run 500 episodes in each experiment. We use ensemble to increase robustness only for the top-down observation. The visualization result of the graph is provided in the Figure 2(b). Despite very few missing edges

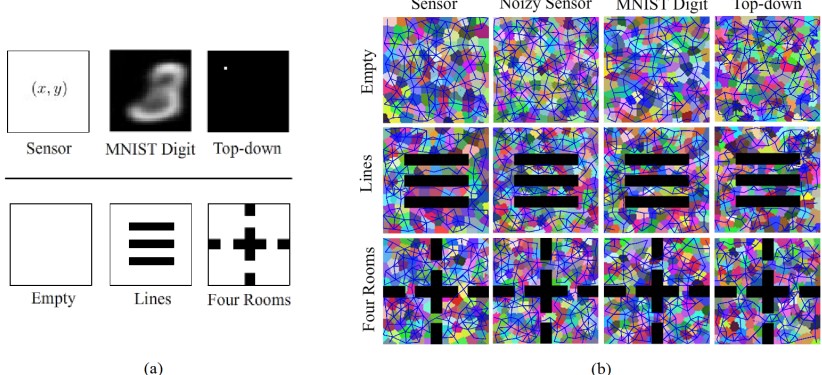

Figure 2: (a) 2D world observations (above) and maps (below). (b) The generated graph of different 2D world environments under different types of observation. Each connected colored segment indicates a vertex with its state coverage. The blue line connecting two segments denotes an edge connecting the corresponding two vertices.

Table 1: Performance of different graph generation algorithms

| Algorithm | Sensor | | | MNIST Digit | | | Top-down | | |
|---|---|---|---|---|---|---|---|---|---|
| | Suc. | Size | Time | Suc. | Size | Time | Suc. | Size | Time |
| SPTM | 73.4% | 10k | > 1s | 60.3% | 10k | > 1s | 61.5% | 10k | > 1s |
| SoRB | 77.6% | 1k | > 0.5s | 56.3% | 1k | > 0.5s | 52.0% | 1k | > 0.5s |
| TOMA | **87.5%** | **0.1k** | **< 0.1s** | **77.2%** | **0.1k** | **< 0.1s** | **75.3%** | **0.1k** | **< 0.1s** |

or wrong edges, the generated graphs are reasonable in 12 cases. The successful result on various observation domains suggest that TOMA is a reliable and robust abstract graph generation algorithm.

### 3.1.2 PLANNING PERFORMANCE

Since learning-based graph generation methods are better than those prior-based methods (Metzen, 2013), we only compare TOMA with recent state-of-the-art learning-based baselines including SPTM and SoRB. We first pretrain a goal-conditioned agent which can reach nearby goals. Then we use the generated graph of TOMA in Section 3.1.1 and SPTM and SoRB to plan for agent respectively. We randomly generate goal-reaching tasks in Four Rooms on three different types of observation. Table 1 shows the average success rate of navigation, the size of the generated graph and the planning time. The agent using the graph of TOMA has a higher success rate in navigation. The main reason behind this is that the generated graph of TOMA can capture more robust topological structure. SPTM and SoRB maintain too many vertices and as a result, we find that they usually miss neighborhood edges or introduce false edges since the learned model is not accurate on all the vertex pairs. Moreover, TOMA also consumes less memory and plans faster compared with other methods. To localize the agent, TOMA only needs to call the embedding network once, and uses the efficient nearest neighbor search to find out the corresponding vertex in $O(\log |V|)$ time. Since TOMA maintains less vertices and edges, the Dijkstra algorithm applied in planning also returns the shortest path faster. The efficiency of planning is crucial since it significantly reduces the training time of Algorithm 2, which requires iterative planning during online sampling.

### 3.2 UNSUPERVISED EXPLORATION

### 3.2.1 SETTING

In this section, we test whether TOMA can explore the sparse-reward environments. The environments for test are MountainCar and another 2D world called Snake maze, which are shown in Figure 3. MountainCar is a classical RL experiment where the agent tries to drive a car up to the hill on the right. Snake maze is a 2D world environment where the agent tries to go from the upper-left corner to the bottom-right corner. In this environment, reaching the end of the maze usually requires 300-400 steps.

In these environments, we set the reward provided by environment to 0. We use DQN (Mnih et al., 2015) as the agent for Mountain-Car and Snake maze as they are tasks with discrete actions. In MountainCar, we set the goal of each episode to be the least visited landmark since the agent needs to explore an acceleration skill and the furthest vertex will sometimes guide the agent into a local minima. In Snake maze, we simply set the goal to be the furthest vertex in the graph. Since HER makes up part of our memory, we use DQN with HER as the baseline for comparison. We test two variants of TOMA: TOMA with vertex memory (TOMA-VM) and TOMA without vertex memory (TOMA). For fair compari-

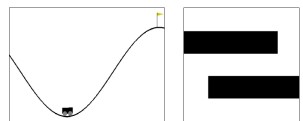

Figure 3: Environments used to test the exploration performance. (Left) Mountain car. (Right) Snake maze.

son, these three methods share the same DQN and HER parameters. For MountainCar, we train the agent for 20 iterations and each iteration lasts for 200 steps. For Snake maze on sensor observation, we train the agent for 300 iterations. For Snake maze on MNIST digit and top-down observation, we train the agent for 500 iterations. Each iteration lasts for 1000 steps. In each iteration, we record the max distance the agent reached in the past history. We additionally calculate a mean reached distance for experiments in Snake maze, which is the average reached distance in the past 10 iterations. We repeat the experiments for 5 times, and report the mean results.

### 3.2.2 RESULT

The results are shown in Figure 5. We can find that TOMA-VM and TOMA outperform the baseline HER in all these experiments. In MountainCar experiment, we find that the HER agent fails to discover the acceleration skill and gets stuck at the local minima. In contrast, both TOMA-VM and TOMA agents can discover the acceleration skill within 3 iterations and successfully climb up to the right hill. Figure 4 (a) shows some intermediate goals of the agent of TOMA-VM, which intuitively demonstrates the effectiveness of TOMA-VM. In Snake maze with sensor observation, the HER agent cannot learn any meaningful action while our TOMA-VM agent can successfully reach the end of the maze. Though the TOMA agent cannot always successfully reach the border of the

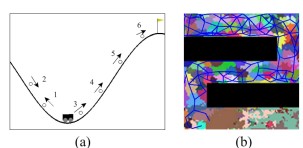

Figure 4: (a) Intermediate goals in MountainCar. (b) The generated graph in top-down Snake maze during exploration.

exploration states in every iteration, there is still over $50\%$ probability of reaching the final goal. In the image based experiments, however, we find that the learning process of goal-conditioned DQN on such a domain is not stable enough. Therefore, our agent can only reach the left or middle bottom corner of the maze on average. A typical example of the generated graph representation during exploration in Top-down Snake maze is shown in Figure 4 (b). This generated graph does provide correct guidance, but the agent struggles to learn the right action across all states. In these experiments, TOMA-VM constantly performs better than TOMA. The reason behind it is discussed in next section.

### 3.2.3 DYNAMICS

We visualize the trajectory and the generated graph during training on the Snake maze with sensor observation in Figure 6. We render the last 10 trajectories and the generated graph every 50 iterations. We find that TOMA will get stuck at the first corner simply because it fails to realize that it should go left, as the past experience from HER pool are mainly for going right and down. In contrast, since TOMA-VM can recall the past experience of reaching the middle of the second corridor, it can successfully go across the second corridor and reach the bottom.

## 4 RELATED WORK

Studies on animals (O'Keefe and Dostrovsky, 1971; Moser et al., 2008; Collett, 1996) reveal that animals are able to build an mental representation to reflect the topological map (graph) of the surrounding environment and aninals will use such representation for navigation. This mental representation is usually termed as *mental map* (Lynch and for Urban Studies, 1960) or *cognitive map* (Tolman, 1948). Furthermore, there exists evidence (Gillner and Mallot, 1998; Driscoll et al., 2000) showing that the mental representation is based on landmarks, which serve as an abstraction of the real environment.

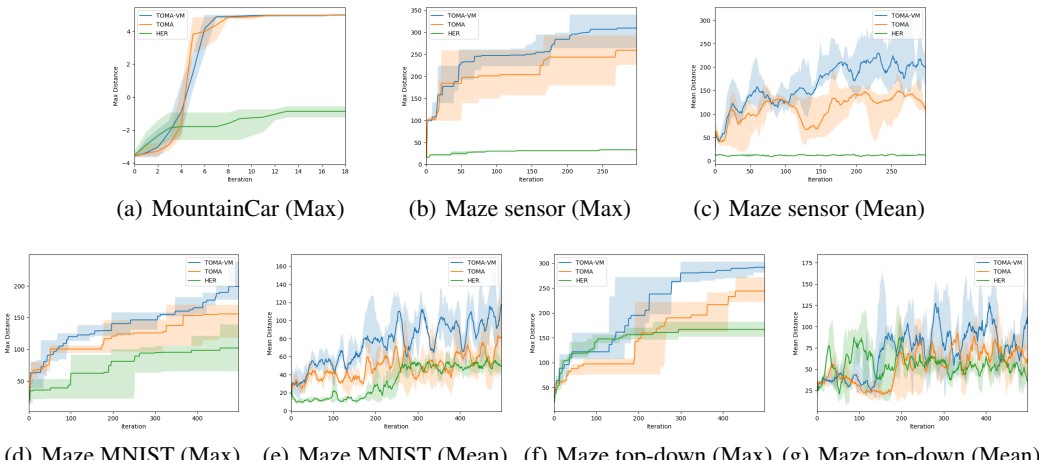

(a) MountainCar (Max)     (b) Maze sensor (Max)     (c) Maze sensor (Mean)

(d) Maze MNIST (Max)    (e) Maze MNIST (Mean)    (f) Maze top-down (Max)    (g) Maze top-down (Mean)

Figure 5: Reached distance of TOMA-VM, TOMA and HER. We also plot the mean of the reached distance for Snake maze experiments. TOMA-VM consistently performs better than the baseline method HER which gets stuck at local minima due to the lack of graph guidance.

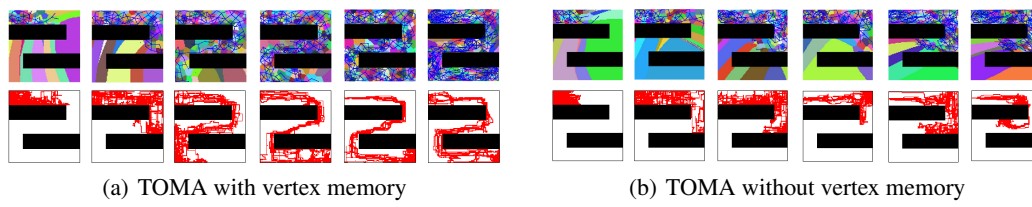

(a) TOMA with vertex memory         (b) TOMA without vertex memory

Figure 6: The generated graphs and last 10 trajectories of TOMA-VM and TOMA every 50 iterations.

Graph is a natural implementation of this mental representation. Researchers have proposed learning to generate graph representation for RL. Existing methods such as SPTM (Savinov et al., 2018) and SoRB (Eysenbach et al., 2019) propose to generate graph representations for planning and they treat the states in the replay buffer as vertices. SPTM learns a reachability predictor and a locomotion model from random samples by self-supervised learning and it applies them over a replay buffer of human experience to compute paths towards goals. SoRB considers the value function of goal-conditioned policy as a distance metric and use it to determine edges between vertices. SoRB requires to train the agent on several random generated goal reaching tasks in the environment during learning. Compared with these approaches which do not adopt abstraction, TOMA generates an abstract graph which has less memory and computation cost and can increase the planning performance. Also, since TOMA is free from constraints such as human experience and training another RL agent, it can be used in RL exploration.

Graph generation methods are related to model-based RL methods (Sutton, 1990) which plan in latent space (Hafner et al., 2019; Kurutach et al., 2018). Recent research (Kara et al., 2020) suggests that model-based RL methods in latent space are hard to train in practice and such drawback can be overcome by using a graph. TOMA is also related to *state abstraction* methods like (Sutton et al., 1999; Singh et al., 1994; Andre and Russell, 2002; Mannor et al., 2004; Ferns et al., 2004; Li et al., 2006; Abel et al., 2016; Roderick et al., 2018) but these methods are prior-based. (Metzen, 2013) points out that learning-based methods perform better than these prior-based methods.

## 5 CONCLUSION

In this paper, we propose a novel graph generation method called TOMA for reinforcement learning, which can generate an abstract graph representation for MDP. Experimental results show that TOMA costs much less memory and computation cost than existing methods and can be used for facilitating exploration. In the future, we will further explore other potential applications of TOMA.

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
