# OpenReview forum: "TOMA: Topological Map Abstraction for Reinforcement Learning"
_ICLR.cc/2021/Conference — Reject_

### Official Review · AnonReviewer2 · 2020-10-19
**Review of "TOMA: Topological Map Abstraction for Reinforcement Learning"**

**Rating:** 4
**Confidence:** 4

**Review:**

**Summary**: This paper studies nonparametric planning in RL, building on prior work in two ways. First, the proposed method creates graph vertices corresponding to *clusters* of observations, rather than single observations. The motivation for this decision is that it leads to sparser graphs, making planning easier. The second idea is to use the graph for exploration by sampling goal states that are near the "fringe" of the graph. Experimental results suggest that the first idea increases performance (success rate and wall-clock time) on 2D navigation tasks. Additional experiments show that the second idea improves exploration.

**Significance**: Nonparametric planning methods have gained traction in the RL community in recent years, and hold the promise of bridging the RL community with the motion planning community. The problems that this paper addresses, increasing the robustness/efficiency of graph construction and improving exploration, are indeed important problems to solve in this area. While this paper makes some progress on these problems, it's unclear how big the improvement is (especially compared with other recent methods, see below), and whether this improvement transfers to tasks beyond 2D navigation.

**Novelty**: Both of the problems that this paper aims to address (building a better graph and using the graph for exploration) have been studied in recent papers. For graph construction, see "Hallucinative Topological Memory for Zero-Shot Visual Planning" [Liu 2020] and "Sparse Graphical Memory for Robust Planning" [Laskin 2020]. For exploration, see "Episodic Curiosity through Reachability" [Savinov 2018], "Skew-Fit: State-Covering Self-Supervised Reinforcement Learning" [Pong 2019], and "Maximum Entropy Gain Exploration" [Pitis 2020]. The paper does not discuss these (very related) papers in any depth, nor does it compare against any of them in the experiments.

**Experiments**: The experimental setup seems reasonable. I especially appreciated that a variety of environment layouts and observation types were used. For the experiments in S3.1, it makes sense that SPTM and SoRB require more time, but it isn't clear to me why they should have lower success rates. In addition, I'd recommend comparing against [Liu 2020] or [Laskin 2020]. It'd be great to include some analysis, visualizations, or ablation experiments to understand why these baselines have lower success rates. For the experiments in S3.2, the plots of the intermediate goals (Fig 4) were quite nice. I'd recommend adding comparisons with at least one exploration bonus method (e.g., RND [Burda 2018]) and one goal-sampling exploration method (e.g., [Savinov 18, Pong 19, Pitis 20]).

**Clarity**: The writing clarity of the paper could be improved, especially in the sections describing the method. See "Minor comments" below.

Overall, I give this paper a score of 4 / 10, primarily because of a lack of comparisons with prior work. I would consider increasing my score if the comparisons suggested above were added. Improving the writing clarity would also help convince me to increase my score.

**Questions for discussion**:
* How are goals sampled for transitions from the vertex memory, and which transitions are used for HER?
* "To increase robustness, we can also use an ensemble ... TOMA does not require ensemble to increase robustness" -- So, does the method use ensembles or not? Do the SoRB and SPTM baselines use ensembles?
* Table 1: Does "size" refer to the number of edges or vertices? What *exactly* does "time" correspond to?
*  In S3.1, do you have any intuition for why SPTM and SoRB should have lower success rates than the proposed method?
* S3.2.1: Why were different environments used in this experiment, as compared with the experiment in S3.1?
* "TOMA is free from constraints such as human experience and training another RL agent" -- In what way do prior methods (e.g., SPTM, SoRB) depend on human experience? It's not clear to me why any of these methods require training another RL agent for exploration.
* "Costs much less memory" -- Where is this shown? (I believe that it's true.)

**Minor comments**:
* There are a number of grammar errors. I'd recommend putting the paper through a grammar checker.
* "This drawback also makes generated graph non-robust" -- Please clarify why nonparametric representations are less robust.
* "existing methods cannot be used for facilitating exploration" -- Can this be clarified?
* "assumes knowledge of the environment" -- SoRB and SPTM *don't* require knowledge of the transition function.
* "This practice is also intractable in RL exploration." -- Exactly what aspect is intractable?
* S2.1: Generally avoid starting sentences with inline math.
* S2.2.1: It's unclear if distance in this section refer to shortest path distances, random walk distances, Euclidean distances, or something else.
* S2.2.1: Throughout this section it's unclear what exactly being learned (I'm assuming it's the parameters $\theta$ of the embedding function $\phi_\theta$). I'd recommend stating at the start what is going to be learned. E.g., "The aim of this section is to learn a latent-space representation of observations $x$ s.t. distances in latent space reflect the Euclidean(?) distance in observation. To this end, we will learn an embedding function $\phi_\theta(x)$ with parameters $\theta$..."
* Eq 1 - 6: I think it'd be helpful to include a plot illustrating these equations.
* "neighborhood biased function $P_s$" -- Where is $P_s$ defined?
* "Anti-Bump Functions" -- This discussion doesn't seem particularly important, and could be cut or moved to the appendix.
* S2.2.2 on the graph generation is pretty hard to follow. Renaming "checking" -> "pruning" might help, as might removing some of the inline math from the three boxes and putting it in the main text (outside the boxes).
* Vertex Memory -- I found this part hard to follow, partially because the method for training the policy hasn't been introduced yet. Also, L6 of Alg 2 seems to have a "type error," as the vertex memory contains "transitions" but HER is a method for sampling "goals."
* "MNIST digit" -- Which digits are shown in which states?

----------------------------------------
**Update after author response**: I appreciate that the authors answered some of my questions, but they did not address my two main concerns: insufficient baselines and writing clarity. I therefore am inclined to maintain my vote to reject the paper (score = 4).

---

> ### Author Response · Authors · 2020-11-24
> **Response to Reviewer 2**
>
> Thank you for your comments. We provide some answers for your questions in the following.
>
> Q1. Does the method use ensembles or not? Do the SoRB and SPTM baselines use ensembles?
>
> We only use ensembles in the Top-down maze. SPTM does not require ensembles. SoRB requires ensembles to improve its performance.
>
> Q2. Does "size" refer to the number of edges or vertices? What exactly does "time" correspond to?
>
> 'Size' refers to the number of vertices. 'Time' refers to the time spent on each single planning operation.
>
> Q3. Do you have any intuition for why SPTM and SoRB should have lower success rates than the proposed method?
>
> In the practice we find that the graphs generated by SPTM and SoRB usually contain some wrong edges (wormholes). In some cases we find that the graph is not even connected. These observations can explain why they have lower success rate.
>
> Q4. In what way do prior methods (e.g., SPTM, SoRB) depend on human experience?
>
> SPTM learns a metric by using random agent and uses this metric to generate the graph over human exploration experience in the experiments. SoRB does not depend on human experience, but it needs to spawn the agent in the environment randomly.
>
> Q5. How are goals sampled for transitions from the vertex memory?
>
> Suppose that we have sampled a sequence $s_1, s_2, ..., s_n$ and its corresponding relabeled sequence is $v_{i_1}, v_{i_2}, ..., v_{i_n}$. Then we uniformly sample some $j$ from $\{1, 2,..., n\}$ and set the goal of subsequence $s_{j-T}, s_{j-T+1}, ..., s_{j}$ to $v_{i_j}$. Here $T$ is a hyperparameter and in our experiments we set it to 16. Then such hindsight experience is put into the vertex memory of $v_{i_j}$.

---

> > ### Comment · AnonReviewer1 · 2020-11-24
> > **Comment from AR1**
> >
> > > SPTM learns a distance metric by exploiting the temporal information from human exploration experience.
> >
> > This is incorrect, SPTM learns the metric from random exploration.

---

> > > ### Author Response · Authors · 2020-11-24
> > > **Thanks for your comment**
> > >
> > > Thank you for your comment. This is indeed a mistake and we have just fixed it in above comment. In our experiments we do use random exploration to learn the metric for SPTM.

---

> > ### Comment · AnonReviewer2 · 2020-11-24
> > **Reviewer response**
> >
> > Thanks for answering my questions! The answers help clarify my understanding of the method.
> >
> > In my review, my main concerns were insufficient baselines and writing clarity. Have either of these issues been addressed in a revised version of the paper?

---

### Official Review · AnonReviewer3 · 2020-10-27

**Rating:** 5
**Confidence:** 3

**Review:**

This work presents a method for learning topological representations for MDPs, encoded as a graph, which could then be used to guide exploration/learning via a goal-conditioned RL mechanism.

The basic idea is appealing, as a form of an intermediate method between model-based and model-free RL. Such intermediate methods which are scalable for large problems are, in principle, of great interest to the community. The paper is overall well-written and easy to follow. However, I have some concerns regarding the approach.

* **Planning**: The authors use the term 'planning' but it should be emphasized that the method does not actually facilitate planning in the "full" sense (as it is commonly understood in RL / control), not even in the aggregated/abstracted state-space. The learned graph only represents *possible* transitions, without representing their probabilities (the behavior of TOMA in non-deterministic environments is not discussed) and, more crucially, without modelling the effect of *actions* on the transitions. On its own this is a valid approach (being kind of a "hybrid" between model-based and model-free), but the use of the term 'planning' (starting from the abstract) is misleading. I think the paper would benefit from a discussion of this point, making the modelling choices more clear/explicit.

* **State Embedding**: The loss function used to train the embedding measures distance between (raw) states by how far apart they were visited in sampled trajectories. However this measure could (and probably will) drastically change when the behavioral policy changes. Particularly in more 'constrained' environments, some states might be seen "far apart" when acting in random, whereas they are in fact closer when a better policy is learned. It is not clear how TOMA handles this issue, and this relates once more to the fact that actions are not being modeled (I believe the 'merge vertices' operator is helpful here, but this is just an intuition). The 'vertex-memory' is a heuristic to overcome this limitation, but it has to go through the policy optimization itself (as it is only affecting the "replay" mechanism) which seems wasteful.

* Another issue, somewhat related to the previous one, is that the method aggregates states uniformly (in term of the clustering "resolution"/granularity, regulated by the parameter $r$). In practice, it could be beneficial to encode in such a way that compresses different parts of the space to different degrees. For example, if there are boundaries/walls with narrow doorways, it would be important for the agent to know its location more accurately when next to such a doorway.

* Finally, the loss function for the embedding seems rather arbitrary. How important is the partition to the particular segments of (-inf,1), (1,3), (3,inf)? How sensitive the model is to the choice of the hyperparameters $\lambda_1,\lambda_2$ (Also, the comment about "ensure that our model focuses on the terms with lower variance" is completely unclear to me. Which variance? where does it come from?). The fact that similar (?) technique is used to prove differential geometry theorems is not very helpful, in my eyes, to understand this method in the context of learning and approximation from samples.

* **Prior work**: Overall this work presents a novel contribution in my eyes. But I think the discussion of its place within the context of prior work could be improved. The (non-exhaustive) references here should not be understood as a criticism for lack of novelty, rather as a suggestion for the authors to provide a more detailed context.

* The idea of dynamically increasing/expanding the learned graph such that it represents the topological structure of the data/environment is closely related to some ideas stemming from the classic Self Organized Feature Maps by Kohonen (e.g Kohonen, 1982, Fritzke 1995, Martinetz and Schulten 1991). Perhaps surprisingly, these ideas or methods haven't been extensively explored in the context of RL and planning (although some work has been done, e.g: Karimpanal & Bouffanais *adaptive behavior* 2019, Smith *Neural networks* 2002 , Lo and Ghiassian *arxiv* 2019).

* Guiding exploration by sampling goals from memory which are "far away" from current/initial state resembles the recent work on "Episodic curiosity through reachability" (Savinov et al.  *ICLR* 2019) which is rather relevant for the current paper. Similar ideas have been studied from several perspectives under "goal conditioned RL" in general.

* I think the "biological" motivation is not too helpful for the paper. The first paragraph in the paper is not accurate (and somewhat outdated). The cited papers (and most of the original place-cells and grid-cells literature) which didn't show that animals can "use such representation to reach the food placed in the maze", but rather studied the way in which space is represented. In any regards, the importance of planning can be explained and justified from computational perspectives which are more relevant for the actual content of this work. It would be better, in my view, to discuss the place of the proposed method on the model-free vs. model-based "extremes" of RL.

---

### Official Review · AnonReviewer1 · 2020-10-27
**Test environments/tasks are not persuasive**

**Rating:** 3
**Confidence:** 5

**Review:**

##########################################################################

Summary:

The paper aims at achieving learning-based graph representations of MDPs with the goal of improving results in sparse-reward RL problems via better exploration. Experimens are mostly done on synthetic environments proposed by the paper itself (besides MountainCar environment). Baselines are SPTM, SoRB and HER.


##########################################################################

Reasons for score:

Experimental evidence in the paper is insufficient to estimate the value of the approach.


##########################################################################

Pros:

1. Interesting ideas about planning for exploration and learnable graph representations for MDPs.
2. Visually the paper looks nice.

##########################################################################

Cons:

1. Test environments/tasks don't look persuasive. Why not pick some more established setups from prior work (e.g., SPTM or SoRB or HER papers) and show advantage on their ground? Creating new setups is only justified if the existing ones are not sufficient with respect to the goals of the paper.
2. The main algorithm idea is not clear. There are many components and the description does not comprehensively show how they fit together and what motivated those design choices.
3. Some statements in the paper would profit from softening/more accurate formulation. To give a few examples:
- "SPTM rely on human sampled trajectories to generate the graph, which is infeasible in RL exploration" - collecting human demonstrations/trajectories is indeed costly but it's not infeasible. In fact, it is a common recipe for kick-starting RL which powered the first versions of AlphaGo and AlphaStar.
- "Another drawback is that existing methods cannot be used for facilitating exploration which is important in RL." - it is worth keeping in mind a paper "Episodic Curiosity through Reachability" https://arxiv.org/pdf/1810.02274.pdf. While it doesn't plan for exploration, it still keeps some kind of topological map in memory, although without explicitly storing edges.
- "In cognitive science society, researchers summarize these discoveries in cognitive map theory (Tolman, 1948), which states that animals can extract and code the structure of environment in a compact and abstract map representation." - that paper came many years earlier than those previously mentioned, so it can't possibly summarize them. In fact, O'Keefe was sceptical about cognitive maps from what I read in those papers.

##########################################################################

Questions during rebuttal period:

Unfortunately, to make paper meet high ICLR standards would require too large a change in my opinion. I would encourage the authors to fix the cons mentioned above and resubmit.

#########################################################################

Minor suggestions and typos:

(1) "which mark three disjoint regions [0; 1]; (1; 3]; (3;+1)," - how did the magical constant 3 come to be?

---

### Official Review · AnonReviewer4 · 2020-10-28
**Interesting approach for learning how to generate RL environment graphs, adequate but relatively simple experiments, writing unclear in some places.**

**Rating:** 5
**Confidence:** 3

**Review:**

This paper proposes a new method for learning graph representations of RL problems. The proposed method, named TOMA, creates a set of *landmarks*, each of which is a representatives state associated with a cluster of states in the original problem.  The clustering is induced via a locality sensitive embedding function. With this embedding, a graph of landmarks is created by sampling state trajectories, and then mapping each state to existing vertices (if a nearby vertex already exist), or creating a new vertex otherwise. To add robustness, an ensemble of embedding functions is used rather than a single estimator. Once the graph is build, the shortest-path to goals can be computed using a search algorithm (e.g., Dijkstra's algorithm). The proposed approach is evaluated on a set of benchmarks, and compares favorably in this problem with recent state-of-the-art graph learning algorithms for RL.

In terms of pros, the proposed approach is interesting and novel, to the best of my knowledge. It presents a suitable addition to recent approaches exploring connections between graph-search and reinforcement learning, as I'm not aware of this particular embedding function being proposed before. I did note that the embedding bears some resemblance to that used in Eq. (5) of Plan2Vec, Yang. et al. 2020 (which is not discussed in the paper, although the paper is cited). Another useful contribution of this paper is the simple exploration technique proposed, of always trying to reach the least explored vertex, and adding a per vertex memory for the agent to keep knowledge of how to reach the graph border. Finally, the experiments illustrate the ability of the approach to construct sound graph representations of the environment, even when given high dimensional noisy observations.

In terms of cons, I think the writing can be improved in some places. For example, the locality sensitive embedding is introduced without explaining any high level intuitions about why this particular embedding should be preferred over others. Another example, is the lack of explanation for case (3) in A1 (why a state d > 3 cannot be a candidate?). Additionally, the environments considered, even accounting for high-dimensional observations, seem relatively simple (SORB considers, for instance, a visual navigation task from RGB images of 3D houses).

Finally, there are some missing references:

- An important one is *Huang, Z., Liu, F., & Su, H. (2019). Mapping state space using landmarks for universal goal reaching. In Advances in Neural Information Processing Systems (pp. 1942-1952).*, which samples far away landmarks (states) using either the euclidean distance, or an estimate provided by a learned value function.

- Another relevant reference is *Corneil, D., Gerstner, W., & Brea, J. (2018, July). Efficient Model-Based Deep Reinforcement Learning with Variational State Tabulation. In International Conference on Machine Learning (pp. 1049-1058)*. This paper builds a discrete representation of the environment (akin to a graph), which is solved using prioritized sweeping.

Overall, I'm inclined towards acceptance, but it would be useful if the authors could clarify some of the aforementioned points, and explain how this approach differs from the references mentioned above.

Minor issues:

- "noizy" --> noisy
- The left panel of Fig. 2 doesn't add much. The bottom left part is redundant (can be seen on right), and the top left could be explained in the text (or as a small vertical column in the image). This would give more space for seeing the generated graph, which I think would be more useful.

---

### Decision · Program_Chairs · 2021-01-07
**Final Decision**

**Decision:**

Reject

**Comment:**

This paper introduces a technique called TOMA to learn abstract graph representations of MDPs. Such an approach is said to be more efficient both in terms of memory and computation. Despite this being an interesting research topic, the reviewers unanimously recommend rejecting the paper. They all agree that the writing needs to be improved for clarity, that some of the algorithmic choices seem arbitrary, and that there are relevant baselines missing. Moreover, the authors didn’t submit a response to most reviewers.